# Comparison of the cloud top heights retrieved from MODIS and AHI satellite data with ground-based Ka-band radar

Juan Huo, Daren Lu, Shu Duan, Yongheng Bi, Bo Liu

Key Laboratory for Atmosphere and Global Environment Observation, Chinese Academy of Sciences, Bejing, 100029, China

*Correspondence to:* Juan Huo (huojuan@mail.iap.ac.cn)

**Abstract.** To better understand the accuracy of cloud top heights (CTHs) derived from passive satellite data, ground-based Ka-band radar measurements from 2016 and 2017 in Beijing are compared with CTH data inferred from the Moderate Resolution Imaging Spectroradiometer (MODIS) and the Advanced Himawari Imager (AHI). Relative to the radar CTHs, the MODIS CTHs are found to be underestimated by on average $-1.10 \pm 2.53$ km and 49% of CTH differences are within 1.0 km.

The AHI CTHs are underestimated by $-1.10 \pm 2.27$ km and 42% are within 1.0 km. Both the MODIS and AHI CTH retrieval accuracy depends strongly on the cloud depth (CD). Large differences are mainly due to the retrieval of thin clouds of CD < 1 km, especially when the cloud base height is higher than 4 km. For clouds with CD > 1 km, the mean CTH difference decreases to $-0.48 \pm 1.70$ km for MODIS and to $-0.76 \pm 1.63$ km for AHI. It is found that MODIS CTHs with higher values (i.e. > 6 km) show smaller discrepancy to radar CTH than those MODIS CTHs with lower values (i.e. < 4km). Statistical

analysis illustrate that the CTH difference between the two satellite instruments is lower than the difference between the satellite instrument and the ground-based Ka-band radar. The monthly accuracy of both CTH retrieval algorithms is investigated and it is found that summer has the smallest retrieval difference.

## 1 Introduction

Clouds play important role in the water and energy budgets in the Earth–atmosphere system (Ramanathan et al., 1989; Liou

1992; Cess et al., 1996; Boucher et al., 2013). They are one of the least-understood components and also one of the largest uncertainty sources in general circulation model (GCM) simulations (Wetherald and Manabe 1988; Arakawa 2004). Cloud top

height (CTH) is one of the important cloud parameters that provide information on the vertical structure of cloud water content (Stubenrauch et al., 1997; Marchand et al., 2010). Cloud vertical distributions determine the diabatic heating profiles. Comparisons of stratocumulus CTHs simulated from GCMs with retrieved from satellite suggested that either satellite retrievals placed stratocumulus clouds too high in the atmosphere or GCMs cloud tops were biased too low (Rossow and Schiffer 1999). Knowledge of CTH is crucial to understand the Earth's radiation budget and global climate change.

Active and passive instruments have long been used for monitoring CTHs (Atlas 1954; Schiffer and Rossow 1983; Pavolonis and Heidinger 2004; Stephens and Kummerow 2007; Huo and Lu 2009; Görsdorf et al., 2015). Active instruments, i.e., cloud radars and lidars, detect CTH directly through reflectivity from cloud top particles. Passive infra-red (IR) instruments measure the IR brightness temperature of the cloud to derive CTH based on assumptions, for instance, that an opaque cloud could be regarded as a black body. Surface measurements and satellite measurements have individual strengths and weaknesses. Some active instruments, i.e., radar, are ideal sensors for accurately detecting the CTH. Yet, surface instruments are limited in spatial scale. Satellites measure large-scale cloud systems, but the CTHs retrieved from passive IR instruments are still subject to large uncertainties. This study assesses the accuracy of the CTHs derived from passive satellite through comparison with surface active radar data.

The Moderate Resolution Imaging Spectroradiometer (MODIS) onboard the Aqua and Terra satellites has been in service since 2000 and the cloud products are being widely used by the meteorological community (King et al., 1998; Ackerman 1998; Rodell and Houser 2004; Roskovensky and Liou 2006; Remer et al., 2008; Pincus et al., 2012). Uncertainties in the MODIS CTH products have been assessed using many measurements, i.e., from the ground, aircraft and satellites (Naud et al., 2002; Weisz et al., 2007; Ham et al., 2009; Chang et al., 2010; Marchand et al., 2010; Baum et al., 2012; Marchand 2013; Xi et al., 2014; Håkansson et al., 2018; Wang et al., 2019). Naud et al. (2002) showed that the two sets of averaged CTHs from the Multi-Angle Imaging Spectroradiometer (MISR) and MODIS were generally within 2 km of each other over the British Isles. Holz et al. (2008) found that MODIS underestimated the CTH relative to the Cloud-Aerosol Lidar with Orthogonal Polarization (CALIOP) by 1.4 ± 2.9 km globally over a two-month period. Xi et al. (2014) found that daytime CTH of marine boundary layer cloud retrieved from MODIS was on average 0.063 km higher than what surface lidar and radar measured. Håkansson et al. (2018) used global Collection-6 MODIS cloud top products to compare with CloudSat-CPR data and

reported that the mean difference was -0.61 ± 2.53 km. Previous global evaluation results might be different to the specific regions. This study compares the retrieved MODIS CTHs with radar measurements in Beijing over a long period and gains further knowledge of the uncertainty of MODIS CTH products.

The Advanced Himawari Imager (AHI) onboard the Himawari-8 (HW8) satellite, a geostationary meteorological satellite, has provided CTHs since July 2015 (Bessho et al., 2016). Zhou et al. (2019) reported that the CTHs derived from surface Ka-band radar from December 2016 to November 2017, were 0.82 km higher than the retrieved CTH from the AHI radiance data using a Fengyun Geostationary Algorithm Testbed-Imager (FYGAT-I) science product algorithm. Mouri et al. (2016) reported that the mean AHI CTH was lower than the MODIS and CALIOP CTH over two weeks of measurements. The AHI CTH retrievals are, relatively, new to the meteorological community and require further evaluation before application in meteorological studies.

MODIS and AHI share part common principles and technologies for the CTH retrieval. However, the specific retrieval algorithms are different in terms of the radiative transfer model, atmospheric profiles, source measurements and cloud types. A Ka-band (35.075 GHz) radar at the Institute of Atmospheric Physics in Beijing, China (39.96°N, 116.37°E) has been used for cloud measurements since 2012 (Huo et al., 2019). In this study, we compare and evaluate the CTHs retrieved from the passive satellite instruments onboard a polar-orbiting satellite and a geostationary satellite with those measured by a surface active radar in Beijing over a long period. To our knowledge, this study presents the first comparison and evaluation of the CTH datasets for Beijing from MODIS and AHI. This work quantifies the satellite CTH retrieval accuracy and provides a reference and usage guidance for the application of the CTH datasets in meteorological research, such as climate model simulations for Beijing and North China.

## 2 Description of the MODIS, AHI and Ka-band radar CTH retrievals

### 2.1 MODIS CTH retrieval

MODIS measures radiance in 36 spectral bands from 0.42 to 14.24 μm at three spatial resolutions: 250 m, 500 m and 1000 m. The swath dimensions are 2330 km (cross-track) by 10 km (along-track at nadir). MODIS cloud top pressure (height) is

determined by a combination algorithm of $CO_2$-slicing technology (also known as the radiance ratioing technology) and infrared-window technology (IRW, using the 11 μm brightness temperature, Smith and Platt 1978; Nieman et al., 1993) in conjunction with the National Centers for Environmental Prediction Global Data Assimilation System temperature profiles (Menzel et al., 2008; Baum et al., 2012). Equations (1)–(3) below present the basic theory of the $CO_2$-slicing technology for which Menzel et al. (2008) presented thorough descriptions.

$$R(v) = (1 - NE)R_{clr}(v) + NE\,[R_{bcd}(v, P_c)], \quad (1)$$

$$R_{bcd}(v,P_c)=R_{clr}(v)-\int_{P_c}^{P_s} \tau(v,p)\frac{dB[v,T(p)]}{dp}\,dp, \quad (2)$$

$$\frac{R(v_1)-R_{clr}(v_1)}{R(v_2)-R_{clr}(v_2)}=\frac{NE_1\int_{P_s}^{P_c}\tau(v_1,p)\frac{dB[v_1,T(p)]}{dp}\,dp}{NE_2\int_{P_s}^{P_c}\tau(v_2,p)\frac{dB[v_2,T(p)]}{dp}\,dp}, (3)$$

where $v$ is the frequency, $E$ is the emissivity of cloud, $R(v)$ is the measured radiance, $R_{clr}$ is the radiance of clear sky, $R_{bcd}$ is the radiance of black body. $N$ is the cloud coverage of the field of view in the range of 0~1, $\tau(v, p)$ is the fractional transmittance of radiation at the wavelength $v$ from the atmospheric pressure level ($p$) arriving at the top of the atmosphere ($p = 0$), $P_c$ is the cloud top pressure, $B[v, T(p)]$ is the Planck radiance at the wavelength $v$ at the temperature $T(p)$, and $P_s$ is the surface pressure. The $CO_2$-slicing technology assumes the emissivity of the cloud to be the same at two close wavelengths,

which is nearly correct for ice clouds, but less so for water clouds.

    For the $CO_2$-slicing technology, Terra-MODIS CTHs are retrieved based on the channels 36/35 and 35/33 (corresponding to 14.2/13.94 μm and 13.94/13.34 μm) ratio pairs due to noise problems at band 34; Aqua-MODIS CTHs are retrieved by the three ratio pairs: channels 36/35, channels 35/34, channels 34/33 (14.2/13.94 μm, 13.94/13.64 μm, 13.64/13.34 μm). In the Collection-5 version algorithm, when the radiance difference between cloud and clear sky is so small

that $CO_2$-slicing technology is unsuitable for CTH retrieval, the IRW is applied. Compared with the Collection-5 version algorithm, Collection-6 differs in terms of the spatial resolution and the radiative transfer model calculation, for example, using ozone profiles provided in the meteorological products rather than from climatological values. Furthermore, application of the $CO_2$-slicing method is limited to only ice cloud in Collection-6 (Baum et al. 2012). The MODIS cloud

products used in this study are the Collection-6 version cloud datasets (MYD06/MOD06) from both Aqua and Terra at 1 km
spatial resolution.

## 2.2 AHI CTH retrieval

The HW8 satellite, equipped with the AHI, was launched on 7 October 2014 at the location of 140.7°E and its operation by
the Japanese Meteorological Agency commenced on 7 July 2015 (Bessho et al. 2016). The AHI is a visible infrared
radiometer that has 16 observation bands, ranging from 0.47 to 13.3 μm (3 for visible, 3 for near-infrared and 10 for infrared).
The AHI observes the Japanese and some other target or landmark areas every 2.5 min and the entire full disk every 10 min
with a spatial resolution of 0.5–2.0 km. The scan ranges for full disk and the Japanese area are preliminarily fixed, while
those for target area and landmark area are flexible to meteorological conditions. Relative to the imagers onboard previous
Japanese geostationary satellites, the AHI is improved in terms of the number of bands, spatial resolution, temporal
frequency and radiometric calibration.

The AHI CTH retrieval algorithm uses radiative transfer codes (Eyre 1991) developed by EUMETSAT and Numerical
Weather Prediction temperature and humidity profile data to calculate the radiance of four infrared bands (wavelengths 6.2,
7.3, 11.2 and 13.3 μm). It involves the interpolation method, the $CO_2$-slicing method and the intercept method. The selection
of method depends on the cloud type from AHI cloud type product (Neiman et al. 1993, Schmetz et al. 1993, Mouri et al.
2016). The interpolation method is similar to the IRW. The vertical profile of radiance at 11.2 μm is calculated using
radiative transfer codes and compared with the radiance observed by AHI. Cloud top height is then determined from the
interpolation ratio of radiance between two levels sandwiching the observed radiance. The interpolation method is adopted
for opaque and fractional cloud. The intercept method uses three scatter diagrams of observed radiances (contain $33 \times 33$
pixels around target pixel) at two band pairs (11.2/6.2 μm, 11.2/7.3 μm and 11.2/13.3 μm) and the calculated black body
radiance curve to determine cloud top height (has the minimum pressure) from the intersection. The intercept method is used
for semi-transparent cloud. In the retrieval process for optically thin (or semi-transparent) clouds, if the intercept method
does not produce suitable results, the $CO_2$-slicing method is applied; if this also fails to produce suitable results, the

interpolation method is utilized. The AHI cloud products used in this study are the Himawari-8 Cloud Property data released through the JAXA's P-Tree System (https://www.eorc.jaxa.jp/ptree/index.html).

## 2.3 Ka-band radar

The Ka-band polarization Doppler radar using a wavelength of 8.55 mm (Ka radar), situated at the Institute of Atmospheric Physics (IAP, 39.967°N, 116.367°E), Beijing, China, was set up in 2010 (Fig. 1). The technical specifications of the Ka-band radar are given in Table 1. The Ka radar works 24 h a day in a vertically pointing mode, except during special events, such as heavy rain or short-term collaborative observations with other instruments, when the mode is changed.

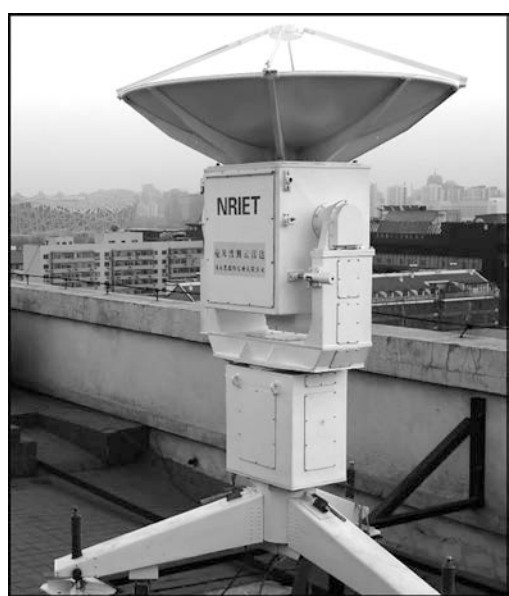

**Fig. 1.** The Ka-band polarization Doppler radar at the Institute of Atmospheric Physics, Chinese Academy of Sciences, Beijing, China (39.967°N, 116.367°E).

A data quality control approach using a combination of the threshold and median filter methods has been implemented to reduce the effects of clutter and noise on the radar reflectivity (Xiao et al. 2018). It is considered to be cloudy if the

125 reflectivity profile contains more than three bins of radar reflectivity data higher than −45 dBZ. Zhou et al. (2019) used a threshold of −40 dBZ for cloud determination because their Ka radar was equipped with an all-solid transmitter which was different to our Ka radar. A higher threshold might miss some clouds with weak returns.

**Table 1.** Main technical specifications of the Ka radar at the IAP.

| | Parameters | Technical Specification |
|---|---|---|
| Transmitter | Frequency | 35.075 GHz |
| | Peak power | 29 kW |
| | Pulse width | 0.2 μs |
| | Type | Magnetron |
| | Pulse repetition frequency | 3.5 kHz |
| Antenna | Diameter | 1.5 m |
| | Gain | 54 dB |
| | Scanning mode | Vertically pointing |
| | Beam width | 0.4 ° |
| Receiver | Noise | 5.8 dB |
| | Noise power | -103 dBm |
| | Vertical resolution | 30 m |

For a cloudy profile, the CTH is determined as the height of the cloudy bin at the highest level. In order to compare with passive satellite data, for clouds detected in a period (i.e., within 10 min), the radar CTH is calculated as the mean CTH of all cloudy profiles, but not upper-level cloud if there are multi-layer clouds. Note that the radar CTH differs when the upper-level cloud covers but not completely covers low-level cloud (see Fig.2). For a cloudy profile, the cloud base height (CBH) is determined by the lowest cloudy radar bin. The cloud depth (CD) is equal to the CTH minus the CBH. The final CBH (or CD) is the average value of all CBHs (or CDs).

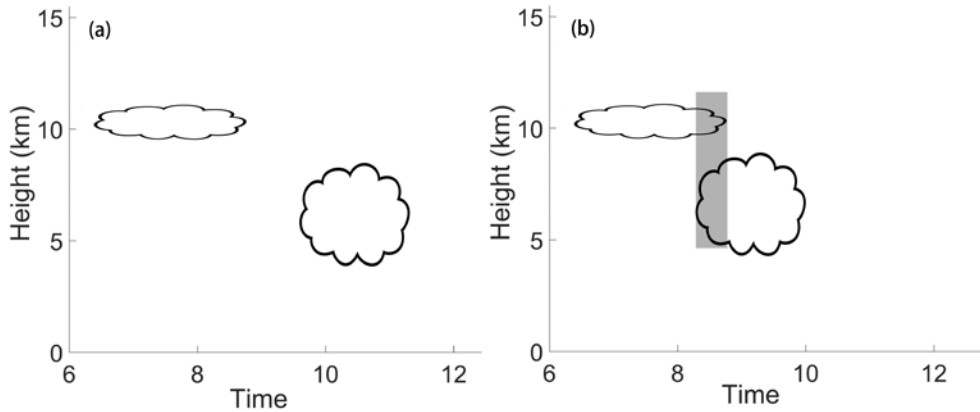

**Fig. 2.** When multi-layer clouds exist, the radar CTH is the mean CTH of all cloudy profiles but not upper-level cloud if (a) upper-level cloud and low-level cloud is situated separately, or if (b) upper-level cloud covers part of the low-level cloud. The CTH of the grey "covered" part of low-level cloud is not considered.

## 3 Comparison scheme

A MODIS or AHI CTH pixel covers larger area than a single profile of radar. The data repetition frequency also differs. Temporal and spatial collocation of the radar, MODIS and AHI data is critical to facilitate effective comparison and evaluation.

### 3.1 Collocation of the MODIS and Ka radar

A MODIS CTH pixel at sub-satellite point covers an area of about 1 km$^2$; vertically pointed radar takes about 1.7 min to scan a 1 km path and about 8 min for a 5 km path if the moving speed of cloud is 10 m·s$^{-1}$. If the moving speed becomes higher (or lower), the required time scanning same path will decrease (or increase). To compensate for the temporal and spatial differences in satellite data and ground-based data, Naud and Muller (2002) used MODIS CTH data averaged over a ±0.1 latitude–longitude box for comparison with surface radar data. Dong et al. (2008) used the surface data (on the Southern Great Plains (SGP) atmospheric observatory established by the Atmospheric Radiation Measurement (ARM)) averaged over a 1 h interval and the satellite data averaged within a 30 km × 30 km area for the surface–satellite comparison. In Holz et al. (2008), the 5 km averaged CALIOP data were collocated with the 1 km MODIS data. These collocation methods are

155 designed to satisfy different instrument and observation conditions.

At the IAP site, a collocation scheme is determined according to the local conditions. The moving speed and direction of cloud is always changing, resulting in variable radar scanning length. The MODIS 1-km spatial resolution is suitable for pixels around the sub-satellite point, but those pixels around IAP site have flexible spatial resolutions due to the viewing geometry of the individual satellite overpasses (see Fig. 3).

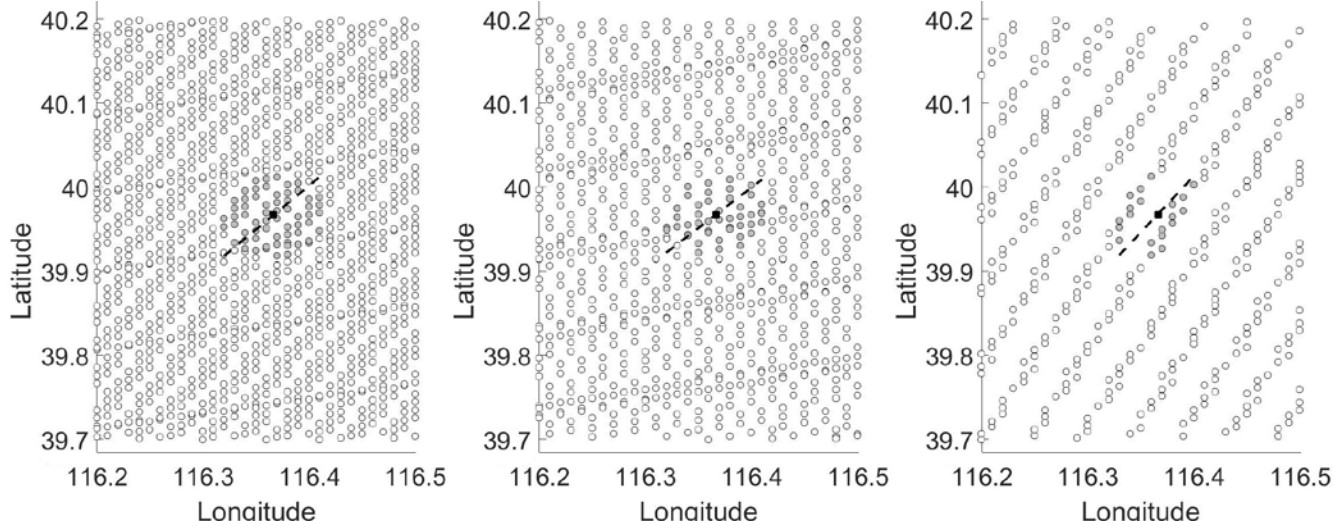

**Fig. 3.** Locations of the Terra MODIS CTHs (circles) and the Ka radar (black solid squares). The three panels exhibit three spatial resolutions of MODIS CTH data due to the changing viewing geometry of the individual satellite overpass. Grey solid circles are MODIS pixels within 5 km to IAP site. The possible path scanned by Ka radar is demonstrated by a black dashed line.

In this study, the ground-based CTH measurements from radar are averaged within 10 min of the MODIS overpass (MODIS observation time ± 5 min). All the MODIS CTHs within 5 km to the IAP site are extracted and averaged to compare with surface radar measurements. A detailed description of the investigation of the optimal collocation scheme, including comparison between collocations using four years of MODIS and radar data averaged on different time and areas, will take place in an additional analysis, but is briefly discussed here. Figure 4 presents the statistics results from four

collocation methods: radar data averaged within 5 min vs. nearest MODIS ($D_{m0-r5}$), radar within 5 min vs. MODIS averaged

on 5 km radius ($D_{m5-r5}$), radar within 5 min vs. MODIS within 30 km ($D_{m5-r30}$), radar within 30 min vs. MODIS within 30 km ($D_{m30-r30}$). It is found that the $D_{m5-r5}$ is close to the average of four collocation methods.

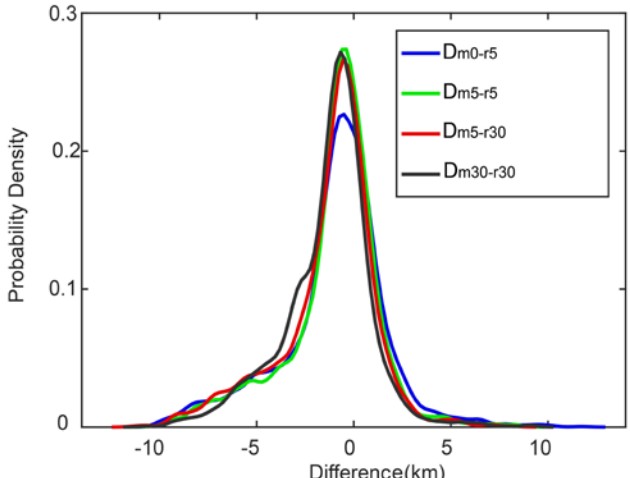

**Fig. 4.** Distributions of the probability density of the CTH difference (MODIS – radar) from four collocation methods. $D_{m0-r5}$: radar data averaged within 5 min vs. nearest MODIS; $D_{m5-r5}$: radar 5 min vs. MODIS averaged within 5 km; $D_{m5-r30}$: radar 5 min vs. MODIS 30 km; $D_{m30-r30}$: radar 30 min vs. MODIS 30 km.

**3.2 Collocation of the AHI and Ka radar**

Due to the Himawari-8 viewing geometry, an AHI CTH pixel has a fixed 5 km×5 km spatial resolution and 10 min temporal resolution over the IAP site (Fig. 5). Since the AHI presents data every 10 min, the Ka radar data within 10 min of the AHI observation time are extracted and averaged (± 5 min). The AHI CTHs nearest to the IAP site are used for comparison.

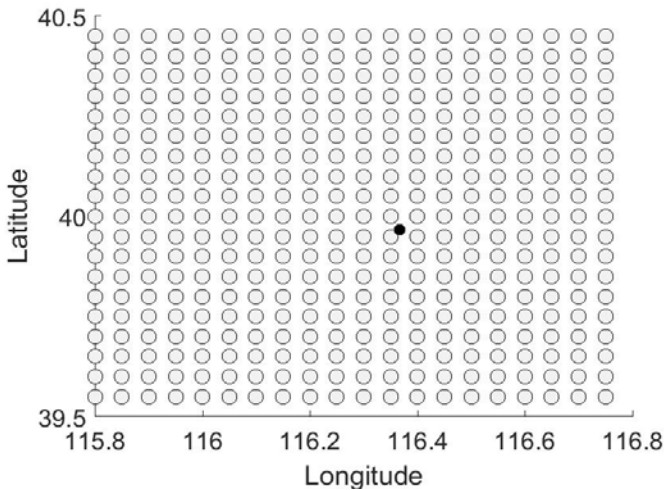

**Fig. 5.** Locations of the AHI CTHs (circles) and the IAP Ka radar (black solid dots). The spatial resolution of the AHI CTH data of full disk does not change.

## 4 Comparison results

In this study, the CTH difference ($D_{mr}$) between the radar and MODIS data, the CTH difference ($D_{ar}$) between the radar and AHI data, and the CTH difference ($D_{am}$) between the AHI and MODIS data is defined as:

$$D_{mr} = H_m - H_r \quad (4)$$

$$D_{ar} = H_a - H_r \quad (5)$$

$$D_{am} = H_a - H_m \quad (6)$$

where $H_r$ is the radar CTH, $H_m$ is the MODIS CTH and $H_a$ is the AHI CTH.

This study uses the radar and satellite data observed from 1 January 2016 to 31 December 2017 to compare.

### 4.1 Comparison between MODIS and Ka radar

After discarding clear-sky data, Ka radar and MODIS have 963 valid CTH comparison pairs from 1 January 2016 to 31 December 2017 (Fig. 6a). The correlation coefficient between MODIS CTHs and radar CTHs is 0.72, which shows a good agreement with each other. Relative to the Ka radar, MODIS tends to underestimate the CTHs, on average by −1.10 ± 2.53 km (mean ± standard deviation (STD) of the $D_{mr}$). Among all comparisons, about 14% differences are within ± 0.25 km, 27% are within ± 0.5 km and 49% are within ± 1.0 km. The statistical result is very close to the global results reported by Håkansson et al. (2018). Figure 6b presents the probability density distribution of the $D_{mr}$ but the peak is not centered at zero.

Most previous comparison studies used the mean bias to describe the CTH difference (Naud et al. 2002; Holz et al. 2008; Chang et al. 2010; Xi et al. 2014). Håkansson et al. (2018) discussed which statistical method was appropriate for describing a non-Gaussian distribution of CTH difference and found the median was better than the mean to describe the tendency. Here, in Fig. 6b, the peak is at −0.30 km (termed "peak" difference) and the median is at −0.57 km with 2.18 km IQR (Inter-Quartile Range). It is clear that the median difference is closer to the peak difference than the mean difference. In this paper, both the mean and median differences are used to describe the CTH difference.

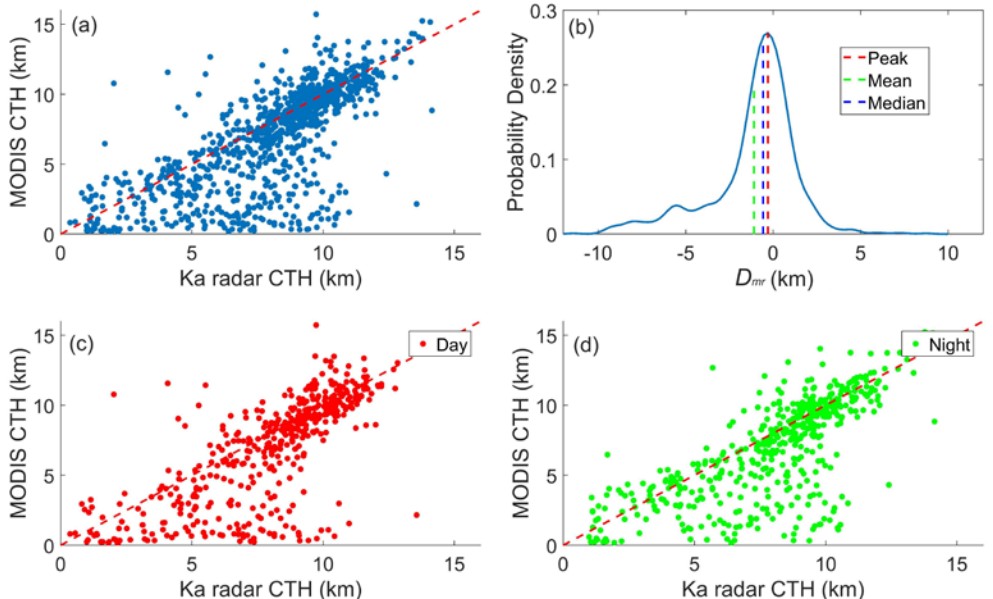

**Fig. 6.** Scatter map of MODIS CTHs and the Ka radar CTHs from (a) all comparisons, (c) comparisons at daytime, and (d) comparisons at nighttime. Panel (b) shows the probability density distribution of the $D_{mr}$, and the peak, mean and median difference is illustrated with red, green and blue dashed line, respectively.

From Fig. 6a, it can be seen that most large-underestimated matches have lower MODIS CTHs, i.e., lower than 4 km. Among all MODIS CTHs, 62% are greater than 6 km of which the mean $D_{mr}$ is 0.0026 ± 1.43 km; yet, the mean $D_{mr}$ is −3.55 ± 2.99 km when the MODIS CTHs are less than 4 km. When compared with lower MODIS CTHs, MODIS CTHs greater than 6 km show better agreement with the Ka radar data. That is, a MODIS CTH greater than 6 km is more likely to be close to radar CTH than a MODIS CTH less than 4 km. Comparisons between day and night show that the mean $D_{mr}$

values during the day and night are close to each other (Fig. 6c, d). Terra MODIS and Aqua MODIS show similar accuracy in the CTH retrieval over Beijing.

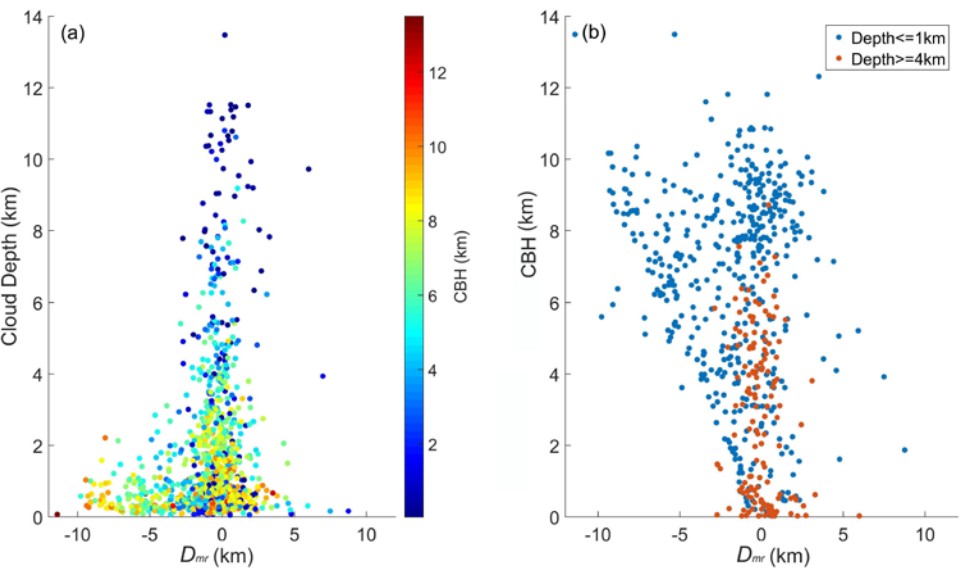

**Fig. 7.** The relationship between (a) the $D_{mr}$ and cloud depth, and (b) the $D_{mr}$ and the cloud base height (CBH) in cases of cloud depths <=1 km (blue) and >=4 km (orange).

Uncertainties in the MODIS CTH retrieval depend strongly on cloud depth (Fig. 7a). When cloud becomes thicker, the range of $D_{mr}$ narrows gradually toward zero. Furthermore, the absolute $D_{mr}$ decreases with increasing CD (Table 2). Large differences are mainly due to thin clouds (CD < 1 km). For clouds with CD > 1 km, the mean $D_{mr}$ is −0.48 ± 1.70 km and the median (IQR) is −0.32 (1.42) km; the mean $D_{mr}$ is −0.29 ± 1.43 km and median (IQR) is −0.29 (0.32) km for CD > 2 km. Figure 7b shows that the $D_{mr}$ changes within small range as the CBH increases when the CD is greater than 4 km. It means that there is no obvious relationship between $D_{mr}$ and CBH for thick clouds. However, for thin clouds of CD < 1 km, MODIS tends to greatly underestimate the CTH of high-level clouds, especially for the clouds with CBH > 4 km. Clouds with CBH > 4 km and CD < 1 km account for 37% of all comparisons, and the mean $D_{mr}$ is −2.16 ± 3.17 km. Clouds of CBH < 4 km and CD < 1 km account for 10% of all cases, and the mean $D_{mr}$ is −0.37 ± 2.07 km. Here, it is found that the MODIS retrieval algorithm shows large uncertainties for high and thin clouds, i.e., CBH is >4 km and CD is <1 km.

**Table 2** The median (IQR), mean and standard deviation of $D_{mr}$ with different CDs (mean ± STD) (unit: km)

| CD | $\in(0,1]$ | $\in(1,2]$ | $\in(2,3]$ | $\in(3,4]$ | $\in(4,5]$ | $>5$ |
|---|---|---|---|---|---|---|
| median(IQR) | −1.14(3.79) | −0.45(1.70) | −0.45(1.38) | −0.28(1.23) | −0.06(1.02) | −0.24(1.43) |
| mean ± STD | −1.74 ± 3.04 | −0.91 ± 2.17 | −0.60 ± 1.61 | −0.18 ± 1.45 | −0.24 ± 1.00 | 0.01 ± 1.30 |

Among all 963 comparisons, 753 comparisons have only one cloud layer. For single-layer clouds, the mean $D_{mr}$ is −1.06 ± 2.39 km and the median (IQR) is -0.55 (1.99) km, while the mean $D_{mr}$ is −1.23 ± 2.98 km and the median (IQR) is −0.70 (2.72) km for multilayer clouds. Cloud occurrence frequency (COF) equals to number of radar cloudy profiles divided by the total number of radar profiles. It is found that the mean $D_{mr}$ declines to −0.39 ± 1.57 km for the comparisons when the CD was >1 km and the COF is >0.5. Here, the MODIS retrieval algorithm shows higher accuracy for continuous clouds than for broken clouds.

**4.2 Comparison between AHI and Ka radar**

Figure 8 shows CTHs from the Ka radar and AHI over 10 h on 9 May 2016. Relative to the comparisons between MODIS and Ka radar, the number of comparisons between the Ka radar and AHI increases due to the increase in temporal resolution of the data. From 1 January 2016 to 31 December 2017, 6719 valid comparisons are found for the CTH comparison.

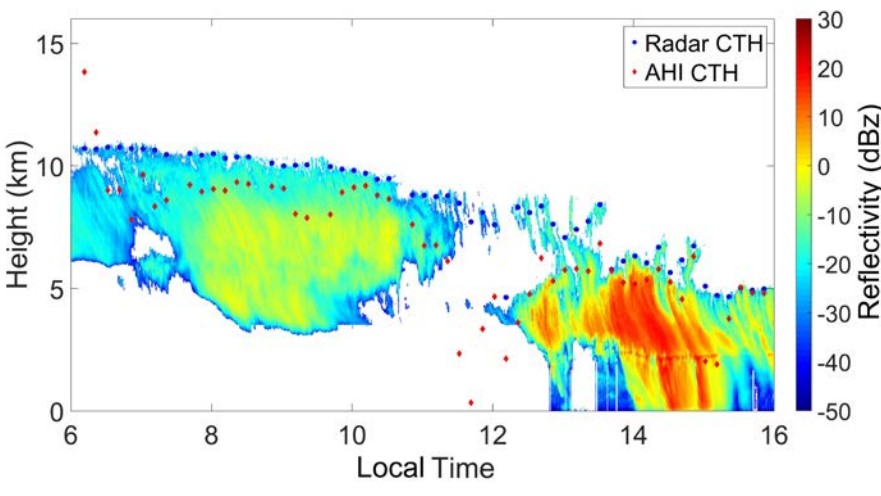

**Fig. 8.** Examples of the Ka radar CTHs (blue dots) and the AHI CTHs (red diamonds) measured on 9 May 2016 from 06:00 to 16:00 (local time: UTC +8).

It can be seen from Fig. 8 that most AHI CTHs are lower than the radar CTHs. All of the 6719 CTH comparisons are shown in Fig. 9a.

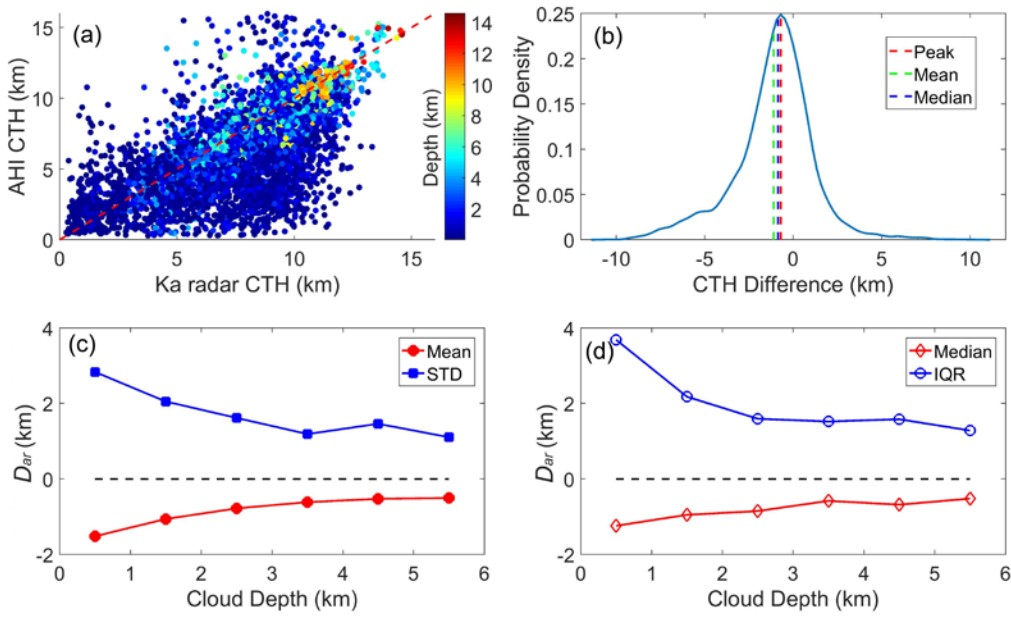

**Fig. 9.** Statistical results of comparisons between Ka radar and AHI. Panel (a) shows the Ka radar CTHs and the AHI CTHs of all comparisons. Panel (b) is same as Fig. 6b but for the $D_{ar}$. (c) The mean $D_{ar}$ and the STD changes with CD. (d) The median of $D_{ar}$ and IQR changes with CD.

Statistically, the mean $D_{ar}$ is $-1.10 \pm 2.27$ km, the median $D_{ar}$ is -0.85 km with IQR 2.32 km and the peak is at -0.70 km (Fig. 9b). About 11% of the differences are less than 0.25 km, 22% are within and 0.5 km and 42% are within 1.0 km. The mean $D_{ar}$ is close to the mean $D_{mr}$. The standard deviation is lower due to more comparisons. However, the median $D_{ar}$ and "peak" $D_{ar}$ is lower than the median $D_{mr}$ and "peak" $D_{mr}$, respectively.

Cloud depth is also a critical factor impacting the accuracy of the AHI retrieval algorithm (Fig. 9c, d). The mean $D_{ar}$ decreases as the CD increases, i.e., the mean $D_{ar}$ is $-1.52 \pm 2.84$ for CD < 1 km while the $D_{ar}$ declines to $-0.76 \pm 1.63$ km for CD > 1 km. The AHI CTHs illustrate great variations for thin clouds (CD < 1 km). The mean bias is larger than the

median bias when CD < 2 km. The relationship between the $D_{ar}$ and the cloud optical thickness (COT) is compared with that between the $D_{ar}$ and CD (Fig. 10). All COTs are from AHI cloud products. The range of the $D_{ar}$ narrows as the COT increases, but the distribution is much more scattered than that for CD, which might be due to the COT retrieval errors.

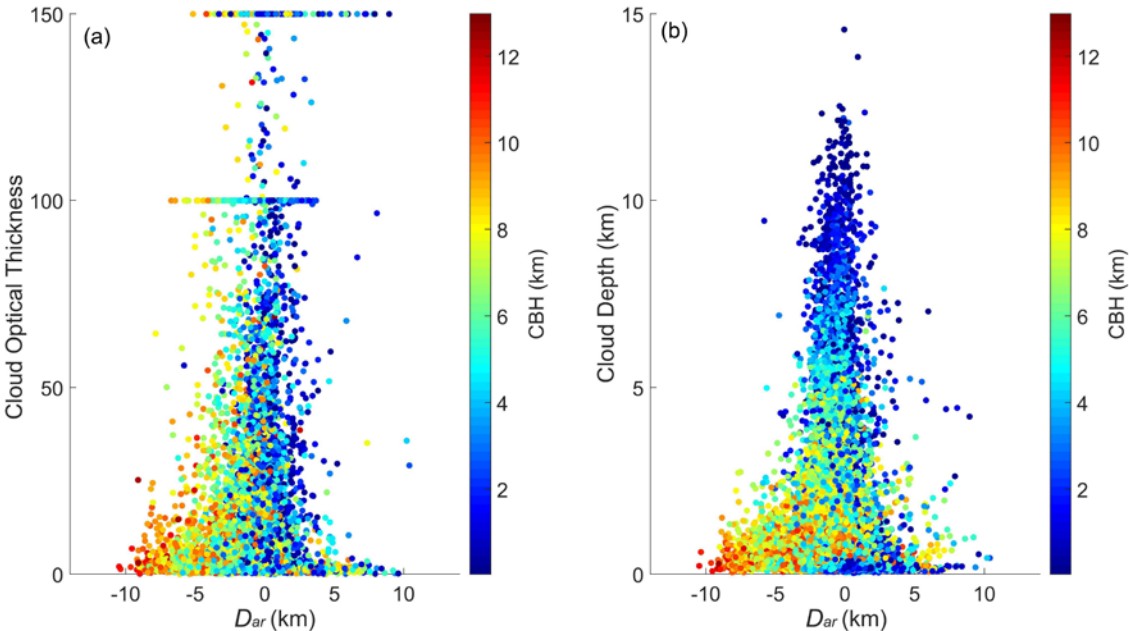

**Fig. 10.** Scatter map illustrates the relationship of (a) the $D_{ar}$ and the COTs, and (b) the $D_{ar}$ and the CD.

Among all comparisons, 79% have only one cloud layer. For single-layer clouds, the mean $D_{ar}$ is −1.12 ± 2.25 km and the median $D_{ar}$ is -0.88 km with IQR 2.25. For multilayer clouds, the mean $D_{ar}$ is −0.99 ± 2.40 km and the median $D_{ar}$ is -0.73 km with IQR 2.69. The impact of COF on the retrieval accuracy cannot be determined because most of the comparisons with CD > 1 km also have COF greater than 0.5.

**4.3 Comparison between MODIS and AHI**

As just addressed, MODIS and AHI CTH data have different spatial and temporal resolutions. This section compares the MODIS CTHs, averaged on data within 2.5 km and 5 km to IAP site, with the nearest AHI CTHs, respectively. Observation time interval of comparisons is limited within 5 min. More than 600 valid comparisons are matched and are shown in Fig. 11.

The mean $D_{am}$ is −0.70 ± 2.49 km for the 2.5 km collocation and −0.64 ± 2.36 km for the 5 km collocation. The median

(IQR) $D_{am}$ is −0.45 (2.18) km for the 2.5 km collocation and −0.43 (1.93) km for the 5 km collocation. The "peak" $D_{am}$ is at

−0.17 km and −0.18 km, respectively. Statistically, the 5 km collocation method shows larger CTH bias than the 2.5 km

collocation. These results are close to the results from Kouki et al. (2016), who reported that the mean AHI CTH was smaller

than the MODIS CTH by −0.54 km based on measurements over 13 days in August. Also, the CTH difference shows an

obvious relationship with COT.

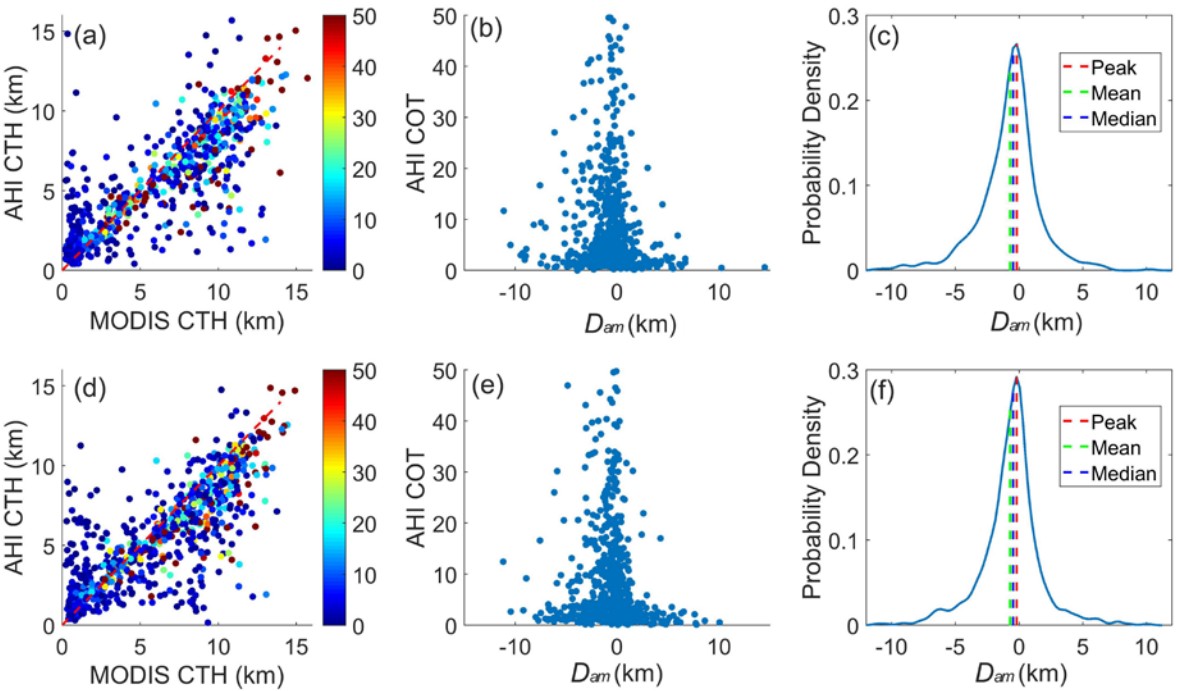

**Fig. 11.** Statistical results of the MODIS CTHs and the nearest AHI CTHs of all comparisons. For the 2.5 km collocation, panel (a) is the scatter map of all MODIS CTHs and the AHI CTHs. Panel (b) shows the relationship between the $D_{am}$ and the COT. Panel (c) presents the probability density distribution of $D_{am}$, same as Fig.9b. Panels (d) - (f) are same as (a) - (c),

but for 5 km radius.

    Based on the analysis in Sections 4.1 - 4.3, an overview of the statistical results is presented in Table 3. Statistically,

MODIS CTHs and AHI CTHs are lower than radar CTHs; the median differences are closer to the "peak" differences than

the mean differences due to the non-Gaussian distribution of difference. Note that the comparisons between MOIDS and

Ka-band, between AHI and Ka-band as well as between MODIS and AHI are based on different comparison samples. Then, due to lower median and peak $D_{ar}$, AHI CTHs on average are lower than MODIS CTHs though the mean $D_{mr}$ and mean $D_{ar}$ is close to each other. It also can be found that the CTH difference between the two satellite instruments is smaller than the difference between the satellite instrument and the ground-based radar.

**Table 3** Statistical results of the distribution of $D_{mr}$, $D_{ar}$, $D_{am}$ (unit: km)

|          | Mean | Median | Peak | STD | IQR |
|----------|------|--------|------|-----|-----|
| $D_{mr}$ | -1.1 | -0.57 | -0.3 | 2.53 | 2.18 |
| $D_{ar}$ | -1.1 | -0.85 | -0.7 | 2.27 | 2.32 |
| $D_{am}$ | -0.64 | -0.43 | -0.18 | 2.36 | 1.93 |

### 4.4 Seasonal variation

The monthly mean and median $D_{mr}$ and $D_{ar}$ are calculated and presented in Fig. 12 as a reference for the meteorological application of the CTH datasets.

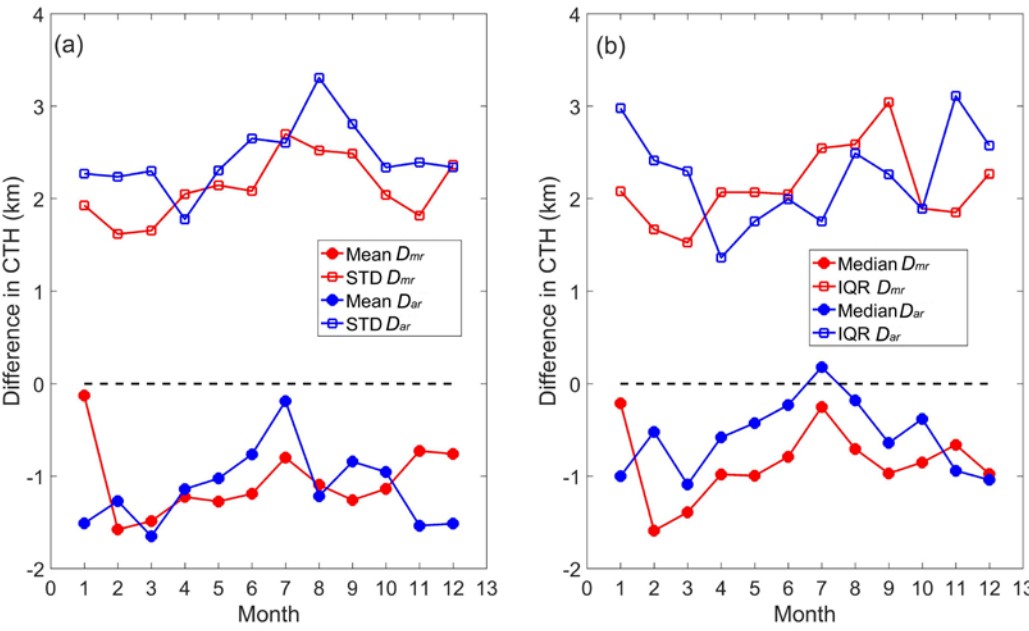

**Fig. 12.** Monthly variations of (a) the mean and standard deviation of the $D_{mr}$ and the $D_{ar}$, and (b) the median and IQR of the $D_{mr}$ and the $D_{ar}$.

Beijing is in North China and has a typical continental monsoon climate. It is located in the subtropical monsoon zone, with southwest and southeast monsoons prevailing in summer and the northwest monsoon prevailing in winter. Rainfall is greater in summer, with less rain but more snow occurring in winter. The cloud distribution also shows strong seasonal variations. As shown in Fig. 12, the monthly variation of mean $D_{ar}$ is greater than mean $D_{mr}$ and shows seasonal characteristics. It is clear that the AHI CTH retrieval algorithm has lower uncertainty in summer (June–August), while it has the largest uncertainty in winter. MODIS CTH retrieval algorithm also shows better performance in summer than other seasons. It is likely associated with the seasonal characteristics of cloud distribution that summer has more thick clouds.

**5 Summaries and discussions**

The accuracy of the CTH retrieval algorithm of MODIS and AHI is associated with the instrument, such as the calibration, signal-noise ratio, the spectral response function and the retrieval algorithm itself, i.e., the atmospheric profile, the calculation accuracy of the radiative transfer model and uncertainty caused by the theoretical assumptions. In an effort to better understand the performance of satellite CTH retrieval algorithms for Beijing, this study evaluates the accuracy of the MODIS and AHI CTH datasets with ground-based radar data based on two years of measurements.

Overall, the CTHs retrieved from the two passive sensors onboard satellites (Aqua/Terra and HW8) are on average lower than the surface radar data. Furthermore, the retrieval accuracy strongly depends on cloud depth. As the retrieval algorithms determine that the retrieved CTH mostly represents the position of the radiation center of the clouds, it is reasonable that most CTHs retrieved by MODIS and AHI are lower than the radar CTHs. The CTH difference between two satellite instruments is smaller than the difference between satellite instrument and ground-based radar.

It is found that retrieved MODIS CTHs greater than 6 km are closer to radar CTH than those lower than 4 km. The large differences are mainly from high and thin clouds (CD < 1 km). In particular, retrieval differences are enlarged when the CBH is greater than 4 km. The mean $D_{mr}$ for clouds with CD > 1 km is −0.48 ± 1.70 km, and it is −0.29 ± 1.43 km for clouds with CD > 2 km. As for the AHI, the mean $D_{ar}$ decreases as CD increases, i.e., the mean $D_{ar}$ is −1.52 ± 2.84 for CD <

1 km while the $D_{ar}$ declines to −0.76 ± 1.63 km for CD > 1 km. The median differences and IQR are also calculated to investigate the CTH difference and it is found that those median CTH biases are smaller than the mean biases due to the non-Gaussian distribution of difference.

Statistical analysis shows that the mean AHI CTHs are lower than the MODIS CTHs over Beijing. On the basis of two years of data, the seasonal changes in the CTH retrieval bias for both sensors is also studied. Both MODIS and AHI retrieval
algorithm have the lowest bias in summer.

This study shows the CTH retrieval accuracy of MODIS and AHI, and provides a reference for better understanding the climatological trends of clouds based on satellite datasets and to enhance their application in GCM models. However, this study does not consider the causes of the retrieval uncertainties. By combining the results of this study with an analysis of the raw radiance data and source retrieval codes, more insights into improvement of the retrieval algorithms can be obtained
in the future.

**Data Availability**

The MODIS product data were obtained from http://ladsweb.nascom.nasa.gov. The AHI data were obtained from https://www.eorc.jaxa.jp/ptree/index.html. The radar data used here are available by special request to the corresponding author (huojuan@mail.iap.ac.cn).

**Author contribution**

Juan Huo and Daren Lu designed the comparisons and Juan Huo carried them out. Shu Duan and Yongheng Bi prepared the ground-based radar data. Bo Liu prepared some Himawari data and references. Juan Huo prepared the manuscript with contributions from all co-authors

**Competing interests**

The authors declare that they have no conflict of interest.

**Acknowledgments**

This work is supported by the National Natural Science Foundation of China (grants 41775032 and 41275040). We appreciate valuable suggestions and insightful instructions from the reviewers. Thanks to the MODIS and the AHI team for sharing their product datasets. We also acknowledge our Ka-radar team for their maintenance service during long-term measurements that made our research possible.

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
