# Peer review of "Comparison of the cloud top heights retrieved from MODIS and AHI satellite data with ground-based Ka-band radar"

_Atmospheric Measurement Techniques, 2019_

## Referee Comment (RC2) · Ulrich Hamann (Referee) · 19 Sep 2019

General comments

The paper of Huo et al compares the cloud top height retrievals of MODIS collection 6, AHI product distributed by JAXA and a ground based Ka-band radar for the region of Beijing. It discuss the differences of the products as function of the cloud base height and cloud depth. The paper is clearly structured and sufficiently well written, but some formulations could be clearer and some typos has to be corrected. These findings are interesting for users of the satellite cloud top height CTH products. The paper would gain a lot, if MODIS and AHI are compared to the Ka-band radar exactly in the same

way (plot Fig 4a and 7a in the same way, plot 4b also for AHI, plot 7b also for MODIS, plot Fig 5 for AHI. . .). The paper would also gain by an extended discussion, e.g. the difference between the radiatively effective CTH (expressed here as "radiation center of the cloud") and the radar CTH was first mentioned in the summary. It might be worth to include a complete discussion chapter. I support that a small summary of the Håkansson et al. 2018 should be included here.

Specific comments

Line 12: Specify here, that you refer to cloud base height: "especially clouds higher than 4km" e.g. write especially when the cloud base height is higher than

Line 12/13/14: Like Nina Håkansson mentioned, these lines should be reformulated, maybe like this: MODIS CTH larger than 6km show a good agreement with the radar CTH when the cloud depth is large and larger differences when the cloud depth is small.

Line 15: I am a bit puzzled by this: The average MODIS CTH is 1.1km lower than the average CTH of the Ka-band. The average AHI CTH is 1.1km lower than the average CTH of the Ka-band. But still the average AHI CTH is 0.64 km lower than the average MODIS CTH. Shouldn't the difference between AHI and MODIS be close to 0 km? Could you comment on this, please?

Line 21: clouds to influence GCM by many more processes, e.g. water transport, radiative transfer, lightning activity, aerosol transport...; maybe write: "for example" as the cloud vertical distribution determine. . .

Line 24: replace "modeled stratocumulus CTHs with satellite retrievals" with "stratocumulus CTHs retrieved from satellite observations"

Line 19 to 24: could be written a bit more smoothly.

Line 30: IR brightness temperature of "the" cloud

Line 30: for instance, "that a" cloud. . .

Line 31: Ground based lidar often does not detect cloud tops.

Line 35 to 45: The references of the MODIS validation that are discussed more in detail are all very old (1999, 2002, 2008). As this paper deals with MODIS collection 6, I suggest to update this section.

Line 51: Mouri et al (2016) found that the CTH (of a ground based radar? where?) was underestimated?

Line 48 / 51: Zhou says that CTH by ground based radar are higher compared to satellite CTH (AHI), but Mouri says CTH (by ground based radar?) was underestimated compared to satellite CTH (MODIS). Could you comment on why the results are different, please.

Line 35 to 57: might be better structured: describe roughly satellite and ground based CTH retrievals, comment on their differences describe global evaluations and describe expected differences (state of the knowledge today); write why local evaluations are important; reason, why this paper brings additional insight; comment on, why the region of Beijing is interesting

Line 71: I suggest to move the equations after the sentence: Equation (1) to (3) present the theory of the CO2-slicing technology.

Equation 1: R_clr must be written in large letters; please explain R_bcd in the text below (I guess this stands for radiance of clouds with emissivity of a black body)

Equation 2 and Equation 3: "dp" is missing at the ends of each integral; You could add a citation here for these equations, e.g. the CLOUD TOP PROPERTIES AND CLOUD PHASE ALGORITHM THEORETICAL BASIS DOCUMENT, Menzel et al.

Line 74: replace "is the radiance measured" with "is the measured radiance"

Line 81: give a few examples of MODIS collection 6 evaluations

Line 83: the spatial resolution (of the cloud product?)

Line 92: target area and landmark areas

Line 95: radiative transfer code (Eyre 1991) developed by EUMETSAT "with" input

Line 104: Avoid the brackets The Ka-band polarization Doppler radar "using a wave-length of 8.55 mm"

Line 133ff: reformulate the last 3 sentences: Zhou used -40 bBZ, but in this study we choose -45 dBZ, as we wanted to include clouds with weak return signals.

Line 127: specify what you mean by "period"

Table 1 add space between number and physical units

Line 126 to 131: this is already comparison technique and should be move the chapter 3

Line 128: I don't understand this sentence: "For multi-layer clouds, the CTH is also the average of all cloudy profiles enven if the upper-level cloud do not cover the lower-level cloud, rather than the average CTH of the upper-level clouds."; Does the upper-level cloud not always cover the lower level cloud by definition? Please comment on, why you choose to average the CTH of the upper cloud layer and the lower cloud layer. (If I understand this correctly.)

Line 131: Could you comment on the accuracy of the Radar CTH measurement, please. Does the radar always sees the uppermost cloud boundary. Or is it possible that attenuation is so strong that you cannot see through the cloud, e.g. during a strong precipitation event.

Line 137: Improve sentence: "MODIS CTH data measured transiently cover an area"

Line 137/138: use superscript for exponents km**2 and m s**-1

Line 138: "line" -> a scanning line with constant elevation

Line 138: what about wind speeds below and above 10 m/s?

Line 147: I would reformulate this: "the MODIS spatial resolution has been increased"; Maybe: depending on the viewing geometry of the individual satellite overpasses, the sampling location and their distances from each other vary, see Fig 2.

Line 148: Please specify more clearly "the climatological distribution of clouds". Do you refer here to the distribution of the cloud movement?

Line 156: Please specify north-south and east-west resolution of the AHI product. Maybe add: "Due to the Himawari-8 viewing geometry", the AHI CTH data have . . .

Line 164: difference between the radar and MODIS (AHI) -> difference between MODIS (AHI) and the radar

Equation 4: write this as two equations to avoid confusion with the notation f(x)=y

Line 173: add plus-minus signs here: less than "+/-"0.25 km, . . . less than "+/-"0.5 km, . . . less than "+/-"1.0 km

Line 175 / Figure 4: I suggest to make Fig 4 and Fig 7 the same, so that it is easy to compare.

Line 191 / Figure 5: In the text, you discuss CTH difference as a function of cloud depth (5a) and CTH difference as a function of CBH. Therefore, I suggest to swap x and y-axis. I also suggest to do a similar figure also for the CTH comparison between AHI and Ka-radar. You might consider to write "D_mr" instead of "CTH Difference"

Line 204: Add more space between Table 2 and Line 204

Line 219++: make a italic "r" in D_ar in this line and all following occurrences

Line 224 / Figure 7: For 7a, use same length for x and y-axis as well as same length. For 7b, you might consider to write "D_ar" instead of "CTH Difference"

Line 231: . . . which might "be" due to

Line 234 / Figure 8: In Fig 8a you might comment on the occurrence peaks of cloud optical thickness = 100 and 150.

Line 250 / Figure 9: You might consider to plot best fit lines in these diagrams.

Line 253-255: The description of the location "subtropical monsoon zone" might already be interesting in the reasoning, why you wrote the paper at the end of the introduction.

Line 265: Here a discussion chapter is missing.

Line 268: uncertainty of the theoretical assumptions -> uncertainty caused by the theoretical assumptions

Line 271: the CTHs retrieved from passive sensors . . . were on average 1.1 km lower compared with the . . .

Line 274: The argument, that the CTH retrieved by satellite

---

## Author Comment (AC1) · 29 Sep 2019

We appreciate Håkansson for her valuable comments and instructions, which helped us to improve the description of results and expressions of the manuscript. Below are the responses to her comments. Modifications in the text are marked with different color. Revised figures, including Figs.2/4/5/7/8/9/10 are pointed out in our responses.

**1    General comments**

The paper compares cloud top height retrievals from two different algorithms on different sensors (MODIS and AHI) with ground based radar data over Beijing. Retrieval accuracy was found to be comparable for the two imagers/algorithms and better for thicker clouds. Results where presented in terms of bias, standard deviation and the percentage of retrieval errors within 0.25, 0.5 and 1km. Overall the paper is well organized and clear but some claims are not supported by the results and references to previous validations of MODIS collection 6 using radar data is missing. Special comment about a missing reference: I am the first author of paper Neural network cloud top pressure and height for MODIS Håkansson et al. 2018: https://www.atmos-meas-tech.net/11/3177/2018/amt-11-3177- 2018-metrics.html. This is a recent paper (2018) evaluating MODIS collection 6 like this paper (not MODIS collection 5 as most of the currently referenced papers) using global space borne CloudSat (CPR) radar data. The results are comparable to the ones in this paper. Further it does discuss which statistical measures are appropriate for describing the resulting non-Gaussian CTH error distributions. I apologize in advance for bringing up my own research but I believe it is very relevant in this case and that this is an important reference that should be included in the article. I am sure the editor or other reviewers can help you decide on inclusion or not, as I am biased.

We are very sorry for missing the reference of Håkansson et al. 2018. Their study is relevant and should be included.

Håkansson et al. (2018) discussed which statistical method was suitable for illustrating the cloud top height difference. Their analysis results presented the instructions how to analyze and show the comparison differences in a more accurate and complete way. According to reviewer's suggestions, we add the analysis of median and IQR differences. Relevant figures and statements are also modified.

**2    Specific comments**

1. Line 12: *"Large differences were mainly occurring for the retrieval of thin clouds of CD < 1 km, especially clouds higher than 4 km"*. seem to be in contradiction to the results at line 13-14: *"MODIS CTHs greater than 6 km showed better agreement with the radar data than those less than 4 km"*. High clouds can not both have better agreement and larger differences? It would be clearer if you add a sentence detailing that radar high thin clouds with large differences will typically have low MODIS CTHs.

   The "CD" and the cloud base height are derived from radar measurement. Only MODIS CTHs are retrieved from MODIS. "High cloud" means cloud has high "radar" base height, but not high MODIS CTH.  There are some high and thin "radar" clouds that have low MODIS CTHs. Just as reviewer had mentioned in the comments 20, "the thin high clouds had a high chance of ending up as MOIDS low clouds".

   To avoid the confusion and make the expression clearer, *"Large differences were mainly occurring for the retrieval of thin clouds of CD < 1 km, especially clouds higher than 4 km" is*

*revised as "Large differences were mainly occurring on the retrieval of thin clouds of CD < 1 km, especially when the cloud base height is higher than 4 km".* The sentence "*MODIS CTHs greater than 6 km showed better agreement with the radar data than those less than 4 km*" is revised as: "*It was found that MODIS CTHs with higher value (i.e. > 6 km) showed smaller difference to radar CTH than those MODIS CTHs with lower value (i.e. < 4km).* "

2. Line 23: *"Statistical analysis showed that the average AHI CTHs were lower than the average MODIS CTHs by −0.64 ± 2.36 km."* For me it was not immediately clear that AHI CTH is statistically significant lower than MODIS CTH given the error distribution. As we are only looking at a sample and the true distribution might very well be centered at zero (as the SD is large and bias is small). Inclusion of a one sample two sided t-test showing that the true bias is not equal to zero would give more support to the claim. However assuming a sample size of 600 the result seem to be significant.

   To inspect what reviewer doubted, we added the probability density of the CTH difference between AHI and MODIS in Fig.9. The CTH differences are given in an added Table 3. The peak CTH difference is -0.18 km and the median is -0.43. It is related with the non-Gaussian distribution of difference what reviewer had discussed in her paper.

3. As seen in figure 4-b the error distributions are non-Gaussian. This makes the interpretation of bias/STD difficult for the reader (see Håkansson et al. 2018 for a longer discussion). Inclusion of medians (or modes) and mean absolute error (or interquartile range) would be helpful. At least medians should be included and discussed.

   We agree with reviewer's suggestions. Only mean (STD) value is inadequate to describe whole CTH difference. We add the median and IQR of all CHT differences analysis which helps readers to get the distribution of difference. Figure 4 and relevant expressions are revised.

4. Line 29: change "is" to "can be" or reformulate to make it clear that the assumption that *"cloud is regarded as black bodies"* is not made by all algorithms. And at least it is not made for all cloud types. Clouds regarded as black bodies are also regarded as opaque. And many algorithms handle also semi-transparent cloud with some skill.

   Sorry for our inaccurate expression. It is revised as "that an opaque cloud can be regarded as a black body".

5. Line 30: Remove the word *"Surface"*. I agree that active sensors are ideal for accurately detection of CTH. Space borne active sensors have the benefit of global coverage. What is the benefit of ground based ones? Is there smaller problem with clutter for ground based radars compared to space borne radars?

   Yes. We agree with reviewer. The "surface" word has been removed.

6. Line 38-39: Many references to MODIS collection 5. Baum et al. is the only one using MODIS collection 6 data and it uses space borne lidar (not radar) for validation. I suggest updating the references list, and at least add more studies of MODIS collection 6 validated with radar data.

   Sorry for our incomplete research investigations of the background. According to reviewer's suggestion, we have made more careful investigations about recent and previous studies and some references are added in the manuscript.

7. Line 40-44: It is better to include validation results for MODIS collection 6 as that is what you are using. And especially as Baum et al. 2012 shows that CTH is much improved in Collection 6 compared to Collection 5.

   According to reviewer's comment, the comparison work for MODIS collection 6 is included in the manuscript.

8. Reformulate/ or remove *"Evaluation results from previous studies are not representative of specific regions"*. The results in global investigations might or might not differ from specific regions, it can not be assumed to be different. It might be different though and that is one reason why your investigation is important.

   It is revised as "Previous global evaluation results might be different to the specific regions".

9. Line 70: A remark: Note that as the CO2 slicing is used only for mid- and high- level clouds, height estimation is needed before the height retrieval can start. A bit of a hen and egg problem.

   It might be better to say that "the $CO_2$ slicing is appropriate only for mid- and high- level clouds". In $CO_2$ slicing technology, the emissivity of the cloud is assumed to be the same at both wavelengths. The assumption is nearly correct for ice clouds and introduces a very small error, but less so for water clouds. We delete this sentence to remove the confusion, add another statement at the end of this paragraph.

10. Line 80: *"Most published evaluation studies on the MODIS cloud top properties are from the Collection-5 version datasets"*. Include and relate your results to some more studies using MODIS Collection 6 CTH data.

    According to reviewer's suggestion, we reformulate the expression. As the purpose of this section is to present a brief description of the MODIS CTH retrieval algorithm, the relevant studies recently published are added in the Introduction.

11. What is the intercept method and what is the interpolation method? Please describe them.

    We add the description of two methods in the manuscript according to reviewer's advice.

12. What method/algorithm/product is used to determine which clouds are semi- transparent?

    The AHI cloud type product indicates the cloud type, which is then used to determine the retrieval method. We have added expressions in the manuscript.

13. Line 107: Are any such *"short-term collaborative observations"* affecting the data used in the study.

    NO. Only vertically pointed Ka-band radar data were used for comparisons. Data from short-term collaborative observations were not included in this study.

14. Line 115: Did you use a lower threshold to include more clouds with weak returns? Or would a lower threshold include more clouds with weak returns that you do not wish to include? Is there a risk for non clouds contamination the results, like aerosols or insects? Please clarify.

    Sorry for our incorrect writing. It should be "*A higher threshold might miss some clouds with weak returns.*"

    The threshold, to some extent, implies the detection ability of radar. Lower threshold indicates stronger detection ability. Our radar can measure very weak returns, like -45 dBz because it uses a magnetron transmitter. A Ka-band radar, equipped with all-solid transmitter,

generally cannot measure weak returns lower than -40 dBz.

There is no risk using lower threshold since the insects generally have strong returns. Aerosols exert main impacts on the visible band but not the Ka band. It is not associated with the weak returns.

Attenuations due to water vapor, oxygen, cloud and precipitation will weaken the returns, and then may underestimate the cloud top height. Thus, if radar transmits weak signal, it is likely to miss some clouds with weaker returns.

15. Table 1. That is a high vertical resolution of 30m! Does this mean that the thinnest clouds detected (three cloudy bins) are only 90m thick? What is the horizontal resolution of the radar data? Please include it in the table.

Yes. Three cloudy bins are 90m. The horizontal resolution of radar data is flexible. It is related with the radar pulse width, pulse repetition frequency, the moving speed of the target, etc. The "Pulse repetition frequency" is added in the table 1. More details about the horizontal resolution are given in our responses to the comment 20.

16. Line127-130: *"For comparison with satellite data, for multilayer clouds in a period, the CTH is also the average CTH of all cloudy profiles even if the upper-level clouds do not cover the lower-level cloud, rather than the average CTH of the upper-level clouds."* This sentence is confusing. Is at any point the CTH of the second level of a multi-layer radar profile included in the averaging? I think the first part of the definition of the radar CTH is very clear as it is. Can the second part be reformulated more like: *Note that for multilayer clouds only the CTH of the highest cloud is used. In the case of a cloudy period/scan with a low cloud which is partly covered by a high second layer cloud the resulting CTH will be an average of the CTH of the upper layer (from the parts with multilayer clouds) and the CTH of the lower layer (from the single-layer parts of the period/scan)*. Or include a figure to make it clear what is done.

Sorry for our poor writing. The cloud top height is not the top height of upper-level cloud. A figure below helps to understand the meaning. In the left figure, final radar CTH is the average CTH of all cloudy profiles of cloud 'a' and cloud 'b' because two clouds are separated. In the right figure, final radar CTH is the average CTH of all cloudy profiles of cloud 'a' and part cloudy profiles of cloud 'b' (cloudy profiles in the gray frame are not included). Note that radar CTH is not the mean CTH of cloud 'a' at the upper level.

[Figure]

These sentence are revised as "In order to compare with satellite data, for clouds detected in a period (i.e. within 5 min or 15 min), the radar CTH is calculated as the mean CTH of all cloudy profiles but not the mean CTH of upper-level cloud if there are multi-layer clouds. That is, the radar CTH might be different to the CTH of upper-level cloud if the upper-level cloud does not cover low-level cloud completely. "

17. Line 133: Suggestion: replace *"data covers larger areas"* to *"have larger field of views"*.

"Field of view" may be inappropriate here because it is related with the viewing geometry of MODIS. We replace "data" with "pixel" and hope reviewer can accept it.

18. Line 134: Reformulate (or at least remove *"Thus"*): *Thus, temporal and spatial collocation of the radar, MODIS and AHI data is critical to facilitate effective comparison and evaluation.* Note that temporal and spatial collocation is necessary regardless of data coverage, FOV (Field Of View) differences, or repetition frequencies. It might be more straightforward in the case of similar FOV and repetition frequencies, though.

    Yes. We agree. "Thus" is removed.

19. Line 148-149: I do not understand the first part of this sentence: *"According to the climatological distribution of clouds, the ground-based CTH measurements from the Ka radar were averaged within 10 min of the MODIS observation time (±5min) in this study"*. Please reformulate.

    This sentence is modified. More details are given in the replies to the following comment.

20. Line 149-150: Note that averaging data in time or space is optional. (In Håkansson et al. we successfully used nearest neighbor matching between CloudSat (CPR) radar data and MODIS collection 6 data). And note that your method, using all MODIS data within 5km and 10min of radar data instead of nearest MODIS 1km pixel and 1.7 minutes of radar data, might decrease the part of imager data actually seen by the radar. The effect would depend on the horizontal resolution of the radar which is unknown to me. A motivation for the averaging of the MODIS data should be included.

    Aqua flies ahead of CloudSat by about 1.5 minutes, like a train. The footprint of CPR is enclosed by the footprint of MODIS. It is appropriate to use the nearest neighbor matching method for the comparison between MODIS and CPR.

    Our Ka-band radar is stationary. In vertically pointed mode, clouds moving into radar's transmitting beam (width $0.4°$) are explored. The length of a scanning "path" by radar is determined by the measuring time, ranges (equals to the height in vertically-pointed mode) and the moving speed of cloud. Then the detecting length, namely horizontal resolution, is flexible. For example, if the speed of a moving cloud is 10 km/h, KPDR takes about 6 min for 1 km and about 30 min for 5 km.

    On the other side, the 1-km spatial resolution of MODIS is appropriate for the data around sub-satellite point. The area of a MODIS pixel is variable and normally increases as the distance to the sub-satellite point increases. As shown in Fig.2, a MODIS pixel covers different areas due to different viewing geometry of MODIS. Then the nearest MODIS data to radar site has various distances. Comparison based on one nearest MODIS pixel will suffer big risk. The nearest neighbor matching method is inappropriate for data collocation of MODIS and ground-based Ka-band radar. Collocation based on more pixels within a certain area and more profiles benefits reducing the risk. That is why there are so many previous works did not use the nearest pixels for the comparison with ground-based radar.

    In fact, we have investigated which time-space collocation scheme is the best for the comparison between MODIS and Ka-band radar. This work has been involved in a submitted paper which now is under reviewing. Here we cite one figure from the paper to illustrate briefly.

    The figure below presents the statistics of the differences using four different collocation methods for the MODIS and radar data: radar 5 min vs. nearest MODIS ($D_{m0\text{-}r5}$), radar 5 min vs. MODIS 5 km ($D_{m5\text{-}r5}$), radar 5 min vs. MODIS 30 km ($D_{m5\text{-}r30}$), radar 30 min vs. MODIS 30 km($D_{m30\text{-}r30}$). It is found that the $D_{m5\text{-}r5}$ is close to the averaged difference of four collocation methods.

[Figure]

In this manuscript, we omit the detailed descriptions of relevant studies considering the sentence balance between AHI and MODIS and hope reviewer can accept.

21. Line 144: *"These collocation methods were designed to match the research goal."* Reformulate to make it sound less as researchers are designing experiments to achieve before-hand determined results. Researchers will make best effort to choose as sound settings as they can. Averaging in time and/or space can improve the study (for example maybe decrease effect of outliers), but can also introduce new problems. For example in situations with clouds of two heights averaging will introduce new types of clouds not present in the original data.

Sorry for that. It is revised as "These collocation methods were designed to satisfy different instrument and observation conditions."

22. In Figure 2: Please note with a different marker (for example x) which MODIS pixels where included in the averaging for each case. Also make the size of the radar dot match the field of view of the radar if it does not already.

Figure 2 has been revised according to reviewer's suggestion.

23. Equation 4: Reformulate or split to two separate equations. Now, because of the parenthesis the reader get the impression the MODIS height $H_m$ is a function of the AHI height $H_a$. A formulation like $D_{mr/ar} = H_{mr/ar} - H_r$ would be better.

The equation has been separated.

24. Line 170: How was the poor quality data defined? Which data were of poor quality MODIS, radar or both? Please clarify.

In fact, the "poor quality" is used to refer to those MODIS data out of valid value range. They have been deleted since the meaning of sentence what we want to tell does not change.

25. Line 173: Nice to see these statistics suitable for non-Gaussian error distribu- tions: *"Among all comparisons, about 14% differences were less than 0.25 km, 27% were less than 0.5 km and 49% were within 1.0 km."* In Håkansson et al. we found for, MODIS collection 6 compared to space borne CloudSat (CPR) *radar data, the part of errors higher than (0.25km, 0.5km and 1km) to be (84%, 70% and 48%). This would correspond to (16%, 30% and 52% of comparisons with in 0.25km, 0.5km and 1km. Considering that our investigation was global and with a space borne radar compared to this investigation using a ground based radarat a single point I think results are noticeably similar.*

Yes. Our results are very close. We add some sentences to state the similarity.

26. Line 181: First I was confused as I did not understand you were splitting results into high low with respect to MODIS CTHs in this sentence. I assumed you would use the radar CTH. But now I understand and the result makes sense as the difficult thin high clouds would have a high chance of ending up as MOIDS low clouds. Please make clearer.

    *Yes. It is what reviewer understood finally. We revised the abstract and this paragraph, and hope the reader can understand it.*

27. Line 205: Cloud occurency frequency. Would *radar cloud fraction* be a better description? The definition is a bit unclear and could be improved. What does cloud time mean? What is observation time? Is it correct that COF = number of radar cloudy profiles divided by the total number of radar profiles (within the 10min time window)?

    *Yes. It is what reviewer means. Sorry for our poor language usage. This sentence has been revised.*

28. Figure7-b: Should it be bias in the legend of the red line? Please include also medians in Figure 7-b.

    *Yes. Sorry for our carelessness. We modify Fig.7, add the figure of the distribution, the median and IQR.*

29. Figure 8 and Figure 9: Please note that it is AHI COTs that are used also in the figures.

    *We add such expression in the caption of Fig.8. We revise Fig.9 according to reviewer's comments here and above.*

30. There are many results in the text; one or more tables giving an overview of the results would help the reader.

    *We add Table 3 in the manuscript to show all quantified difference.*

31. Line 236 and Line 281: *"Statistically, the AHI retrieval algorithm showed better performance for multilayer clouds than single-layer clouds"*. Better performance for multi-layer clouds compared to single-layer clouds is the opposite of what would be expected for any CTH algorithm. This means that strong evidence is needed to support such a claim. Note that bias is better for multilayer clouds but STD is higher. So there is no support for the claim that AHI would retrieve more accurate CTH for multi-layer clouds compared to single-layer clouds. Please reformulate.

    *We delete this sentence according to review's comment. The median and IQR are added.*

32. Line 237: *Compared with $\overline{MODIS}$, the AHI retrieval algorithm showed a slightly better performance for multilayer clouds.* Can these numbers really be used to say one algorithm is better than the other? If I understand correctly, in these investigations it is not only cases where all three instrument match (AHI, MODIS and the radar), instead there is one data set with all MODIS-radar matches and another with all AHI-radar matches. And there are only 210 multi-layer MODIS- radar matches which is a quite small sample size. And this means that the population bias of all multi-layer clouds actually can be quite different from the sample bias (-1.23km) calculated on 210 samples. A difference between sample bias and actual bias in the order of $2.6SD/(210^{1/2}) = 0.5km$ is very well possible. Therefore I think there are not enough results to support the claim that AHI perform better than MODIS for multilayer clouds. In my opinion all three datasets should be matched, or/and the differences between the distributions

(or the number of samples) need to be larger in order to form enough support for the claim. Please reformulate or update with a statistical test supporting the claim.

Yes. The comparison datasets are different. We agree with the reviewer, especially after we calculated the median and IQR of the difference. We deleted this sentence in the manuscript.

33. Line 237-238: Use COF defined on line 205.

It is revisese.

34. Line 241: Is it as in the other investigations: MODIS data are averaged and for AHI the nearest pixel is used?

Yes. Relevant texts are revised to make the meaning clear.

35. Line 243: Please include also median differences.

Median difference has been added.

36. Line 256: As the error distributions are non-Gaussian I would be very cautious to recommend using bias and STD in any meteorological application.

The median and IQR are added.

37. At least add median to Figure 10. The difference between bias/and median at least show if the distributions are skewed. Even if the high kurtosis of the error distributions would still be hidden.

The median and IQR are added in Fig.10.

38. Line 268: Note that all CTH algorithms do not include radiative transfer models. Or maybe they do, at least implicitly? Please motivate or reformulate.

The "passive satellite sensors" are replaced with the "AHI" and "MODIS" , ensuring an accurate statement.

39. Line 272: At least include also median.

We removed the words " by .....km" since mean and median difference are different and they have been presented in Table 3.

40. Line 284: It is not certain that MODIS had the lowest accuracy in spring. It had the lowest bias (but bias is not the same as accuracy, especially not for non–Gaussian distributions!). Note that also standard deviations were low for MODIS in spring which would indicate better accuracy. If you want to use a single measure to evaluate the accuracy of the algorithms I would recommend mean absolute error. Please update.

According to reviewer's comment, this statement is removed.

41. Figures showing the distributions for $D_{ar}$ and AHI-MOIDS should be included. Similar to the one for $D_{mr}$ in Figure 4-b.

The distribution figures have been added in Fig.7 and Fig.9

**3 Technical corrections**

Line 231: "might due to" => "might be due to" or maybe better "could be caused by"

It has been revised.

---

## Author Comment (AC2) · 29 Sep 2019

We deeply thank the reviewer for her/his thoughtful comments and suggestions, which benefited the improvement of this manuscript. Below are the responses to her/his comments.

**General comments**

The paper of Huo et al compares the cloud top height retrievals of MODIS collection 6, AHI product distributed by JAXA and a ground based Ka-band radar for the region of Beijing. It discuss the differences of the products as function of the cloud base height and cloud depth. The paper is clearly structured and sufficiently well written, but some formulations could be clearer and some typos has to be corrected. These findings are interesting for users of the satellite cloud top height CTH products. The paper would gain a lot, if MODIS and AHI are compared to the Ka-band radar exactly in the same way (plot Fig 4a and 7a in the same way, plot 4b also for AHI, plot 7b also for MODIS, plot Fig 5 for AHI. . .). The paper would also gain by an extended discussion, e.g. the difference between the radiatively effective CTH (expressed here as "radiation center of the cloud") and the radar CTH was first mentioned in the summary. It might be worth to include a complete discussion chapter. I support that a small summary of the Håkansson et al. 2018 should be included here.

Thank reviewer. According to both reviewers' suggestions, many modifications have been made in this manuscript and hope the revised manuscript can meet the approval of reviewer.

**Specific comments**

1. Line 12: Specify here, that you refer to cloud base height: "especially clouds higher than 4km" e.g. write especially when the cloud base height is higher than

   According to reviewer's advice, the sentence is revised as "*especially when the cloud base height is higher than 4 km*"

2. Line 12/13/14: Like Nina Håkansson mentioned, these lines should be reformulated, maybe like this: MODIS CTH larger than 6km show a good agreement with the radar CTH when the cloud depth is large and larger differences when the cloud depth is small.

   Radar measures cloud depth and cloud base height. Only the MODIS CTH is from MODIS. Both high MODIS CTHs and low MODIS CTHs are the top heights of clouds with various cloud base heights and cloud depths. It is possible that a low MODIS CTH is retrieved from the cloud with higher base height since the base height is reported by radar. To avoid confusion, this sentence is revised as "It was found that MODIS CTHs with greater value (i.e. > 6 km) showed smaller difference to radar CTH than those MODIS CTHs with lower value (i.e. < 4km)."

3. Line 15: I am a bit puzzled by this: The average MODIS CTH is 1.1km lower than the average CTH of the Ka-band. The average AHI CTH is 1.1km lower than the average CTH of the Ka-band. But still the average AHI CTH is 0.64 km lower than the average MODIS CTH. Shouldn't the difference between AHI and MODIS be close to 0 km? Could you comment on this, please?

   For the comparison MOIDS vs. Ka-band, AHI vs. Ka-band or MODIS vs. AHI, we collocate the CTH data individually. Thus, the three comparison results are based on different samples. It is the main reason why the difference between AHI and MODIS is not close to 0 km. Another reason may be related with the statistical method. The new Table 3 presents an overview of the statistical results in which the median and peak differences of $D_{ar}$ and $D_{mr}$ are different.

*To avoid the confusion, this sentence "Statistical analysis showed that the average AHI CTHs were lower than the average MODIS CTHs by −0.64 ± 2.36 km" is revised as "Statistical analysis showed that the CTH difference between two satellite instruments AHI and MODIS was lower than the difference between satellite instrument and ground-based Ka-band radar.*

4. Line 21: clouds to influence GCM by many more processes, e.g. water transport, radiative transfer, lightning activity, aerosol transport...; maybe write: "for example" as the cloud vertical distribution determine...

   *It has been revised according to reviewer's suggestion.*

5. Line 24: replace "modeled stratocumulus CTHs with satellite retrievals" with "stratocumulus CTHs retrieved from satellite observations"

   *Sorry for our poor writing. Reviewer's advice is different to what we want to express. This sentence is revised as "Comparisons of stratocumulus CTHs simulated by GCMs with retrieved from satellite...."*

6. Line 19 to 24: could be written a bit more smoothly.

   *We have made some minor revisions according to reviewer's advice.*

7. Line 30: IR brightness temperature of "the" cloud

   *"the" is added.*

8. Line 30: for instance, "that a" cloud...

   *It has been revised as "that an opaque cloud can be"*

9. Line 31: Ground based lidar often does not detect cloud tops.

   *"Some" is added.*

10. Line 35 to 45: The references of the MODIS validation that are discussed more in detail are all very old (1999, 2002, 2008). As this paper deals with MODIS collection 6, I suggest to update this section.

    *According to reviewer's suggestion, we add and update recent references in the manuscript.*

11. Line 51: Mouri et al (2016) found that the CTH (of a ground based radar? where?) was underestimated?

    *It is the "AHI CTH". The sentence is revised as "Mouri et al. (2016) reported that the AHI CTH was underestimated compared with the MODIS and CALIOP data over two weeks of measurements". In the study of Mouri et al. (2016), the place is not mentioned.*

12. Line 48 / 51: Zhou says that CTH by ground based radar are higher compared to satellite CTH (AHI), but Mouri says CTH (by ground based radar?) was underestimated compared to satellite CTH (MODIS). Could you comment on why the results are different, please.

    *Sorry for our poor writing, resulting in reviewer's misunderstanding. Zhou et al. compared the CTH retrieved from AHI radiance data using a different CTH retrieval algorithm with the CTH derived from Ka-band radar. Zhou's CTH retrieval algorithm is not the AHI CTH retrieval algorithm. Mouri compared the AHI CTH with the CTH derived from MODIS and CALIOP.*

    *Studies of Zhou and Mouri are the publications what we can find at present about investigating the CTHs derived from AHI measurements. The comparison dataset are different and the retrieval*

algorithm are also different, so the results are different.

13. Line 35 to 57: might be better structured: describe roughly satellite and ground based CTH retrievals, comment on their differences describe global evaluations and describe expected differences (state of the knowledge today); write why local evaluations are important; reason, why this paper brings additional insight; comment on, why the region of Beijing is interesting

    Yes. We revised this section according to both reviewers' comments and hope those revisions can be accepted.

14. Line 71: I suggest to move the equations after the sentence: Equation (1) to (3) present the theory of the CO2-slicing technology. Equation 1: R_clr must be written in large letters; please explain R_bcd in the text below (I guess this stands for radiance of clouds with emissivity of a black body) Equation 2 and Equation 3: "dp" is missing at the ends of each integral; You could add a citation here for these equations, e.g. the CLOUD TOP PROPERTIES AND CLOUD PHASE ALGORITHM THEORETICAL BASIS DOCUMENT, Menzel et al.

    Thank reviewer for his careful reviewing and we are very sorry for our carelessness. Equation 1: $R_{clr}$ is revised in large letter. Explanation of the $R_{bcd}$ is added. "d$p$" is added in Eq.2 and Eq.3.
    The sentence is revised according to reviewer's suggestion.

15. Line 74: replace "is the radiance measured" with "is the measured radiance"

    It is revised.

16. Line 81: give a few examples of MODIS collection 6 evaluations

    This has been added in the introduction.

17. Line 83: the spatial resolution (of the cloud product?)

    Cloud products of Collection 5 and Collection 6 have different spatial resolutions. This sentence is reformulated.

18. Line 92: target area and landmark areas

    It is revised.

19. Line 95: radiative transfer code (Eyre 1991) developed by EUMETSAT "with" input

    It is revised as "The AHI CTH retrieval algorithm uses radiative transfer codes (Eyre 1991) developed by EUMETSAT and Numerical Weather Prediction temperature and humidity profile data to calculate the radiance of four infrared bands (wavelengths 6.2, 7.3, 11.2 and 13.3 μm)."

20. Line 104: Avoid the brackets The Ka-band polarization Doppler radar "using a wave- length of 8.55 mm"

    It is revised according to reviewer's advice.

21. Line 133ff: reformulate the last 3 sentences: Zhou used -40 bBZ, but in this study we choose -45 dBZ, as we wanted to include clouds with weak return signals.
    These sentences are revised according to both reviewers' comments. Ka-band radar equipped with magnetron-type transmitter can transmit stronger signals than the radar with all-solid transmitter. So, it can detect weaker returns.

22. Line 127: specify what you mean by "period"

It is revised as "For clouds detected in a period (i.e. within 5 min or 15 min),...."

23. Table 1 add space between number and physical units
They have been added.

24. Line 126 to 131: this is already comparison technique and should be move the chapter 3
These sentences describe how radar CTH is calculated or derived. It might be better in section 2.3.

25. Line 128: I don't understand this sentence: "For multi-layer clouds, the CTH is also the average of all cloudy profiles even if the upper-level cloud do not cover the lower-level cloud, rather than the average CTH of the upper-level clouds."; Does the upper-level cloud not always cover the lower level cloud by definition? Please comment on, why you choose to average the CTH of the upper cloud layer and the lower cloud layer. (If I understand this correctly.)
Another reviewer also has confusions. Sorry for our poor usage of English language. These sentences are revised. The following figure helps to make it clearer.

[Figure]

In the left figure, final radar CTH is the average CTH of all cloudy profiles of cloud 'a' and cloud 'b'. In the right figure, final radar CTH is the average CTH of all cloudy profiles of cloud 'a' and part cloudy profiles of cloud 'b' (where cloudy profiles in the gray frame are not included). We did not use the mean CTH of the upper-level cloud 'a' as final radar CTH because that might bring more bias to what satellite measures within an area.

26. Line 131: Could you comment on the accuracy of the Radar CTH measurement, please. Does the radar always sees the uppermost cloud boundary. Or is it possible that attenuation is so strong that you cannot see through the cloud, e.g. during a strong precipitation event.
Yes. Ka-band radar has an algorithm to correct the reflectivity attenuation due to water vapor and oxygen but the attenuation by cloud and precipitation has not been calibrated. Attenuations due to cloud and precipitation will affect the determination of cloud top and the cloud top height is probably underestimated. We now cannot tell the accuracy. But the error is expected to be minor since radar has a stronger signal transmitter and its vertically pointing mode limits the attenuation paths, which are should be less than 16 km.

27. Line 137: Improve sentence: "MODIS CTH data measured transiently cover an area"
The sentence is revised.

28. Line 137/138: use superscript for exponents km**2 and m s**-1
They are revised.

29. Line 138: "line" -> a scanning line with constant elevation

It is revised.

30. Line 138: what about wind speeds below and above 10 m/s?

    If the moving speed becomes higher (or lower), the required time for scanning same path will decrease (or increase).

31. Line 147: I would reformulate this: "the MODIS spatial resolution has been increased"; Maybe: depending on the viewing geometry of the individual satellite overpasses, the sampling location and their distances from each other vary, see Fig 2.

    It has been revised according to reviewer's advice.

32. Line 148: Please specify more clearly "the climatological distribution of clouds". Do you refer here to the distribution of the cloud movement?

    Same question is asked by another reviewer. Could you please see my replies to the comments of that reviewer (Comment 20)?

33. Line 156: Please specify north-south and east-west resolution of the AHI product. Maybe add: "Due to the Himawari-8 viewing geometry", the AHI CTH data have . . .

    It has been revised according to reviewer's advice.

34. Line 164: difference between the radar and MODIS (AHI) -> difference between MODIS (AHI) and the radar

    The equations are separated. The sentences are revised according to reviewer's advice.

35. Equation 4: write this as two equations to avoid confusion with the notation f(x)=y

    The equations are separated.

36. Line 173: add plus-minus signs here: less than "+/-"0.25 km, . . . less than "+/-"0.5 km,. . . less than "+/-"1.0 km

    They are revised according to reviewer's advice.

37. Line 175 / Figure 4: I suggest to make Fig 4 and Fig 7 the same, so that it is easy to compare.

    Figure 4 and Fig.7 are revised. Both revised figures present scatter map and the distribution of difference. In Fig.4, data are separated by day and night because they are from two satellites (Terra and Aqua). In Fig.7, it is not necessary to make such separation because data come from one satellite.

38. Line 191 / Figure 5: In the text, you discuss CTH difference as a function of cloud depth (5a) and CTH difference as a function of CBH. Therefore, I suggest to swap x and y-axis. I also suggest to do a similar figure also for the CTH comparison between AHI and Ka-radar. You might consider to write "D_mr" instead of "CTH Difference"

    "CTH difference" is replaced with "$D_{mr}$".

    The impacts of depth on the retrieval algorithm can be seen clearly from current Figure 5. We do not revise them. But if reviewer insist, we can make modification.

39. Line 204: Add more space between Table 2 and Line 204

    They have been added.

40. Line 219++: make a italic "r" in D_ar in this line and all following occurrences
    All have been revised.

41. Line 224 / Figure 7: For 7a, use same length for x and y-axis as well as same length. For 7b, you might consider to write "D_ar" instead of "CTH Difference"
    They have been revised.

42. Line 231: … which might "be" due to
    It has been revised.

43. Line 234 / Figure 8: In Fig 8a you might comment on the occurrence peaks of cloud optical thickness = 100 and 150.
    The COTs are from AHI. Uncertainties of the retrieved COT are not available at present.

44. Line 250 / Figure 9: You might consider to plot best fit lines in these diagrams.
    According to two reviewer's comments, Fig.9 has been revised.

45. Line 253-255: The description of the location "subtropical monsoon zone" might already be interesting in the reasoning why you wrote the paper at the end of the introduction.
    Yes. We reformulate this section to make the expression smoother.

46. Line 265: Here a discussion chapter is missing.
    Sorry. In fact, the summary chapter includes some discussions. We make some minor revisions and change the section title.

47. Line 268: uncertainty of the theoretical assumptions -> uncertainty caused by the theoretical assumptions
    It is added.

48. Line 271: the CTHs retrieved from passive sensors … were on average 1.1 km lower compared with the …
    It is revised according to reviewers' advice.

49. Line 274: The argument, that the CTH retrieved by satellite
    It is revised.